# Beyond Accuracy: Evaluating Self-Consistency of Code Large Language Models with IdentityChain

**Marcus J. Min**[1]    **Yangruibo Ding**[1]    **Luca Buratti**[2]    **Saurabh Pujar**[2]
**Gail Kaiser**[1]    **Suman Jana**[1]    **Baishakhi Ray**[1]
[1]Columbia University    [2]IBM Research
jm5025@columbia.edu
{yrbding,kaiser,suman,rayb}@cs.columbia.edu
{luca.buratti2,saurabh.pujar}@ibm.com

## ABSTRACT

Code Large Language Models (Code LLMs) are being increasingly employed in real-life applications, so evaluating them is critical. While the conventional accuracy evaluates the performance of Code LLMs on a set of individual tasks, their self-consistency across different tasks is overlooked. Intuitively, a trustworthy model should be self-consistent when generating natural language specifications for its own code and generating code for its own specifications. Failure to preserve self-consistency reveals a lack of understanding of the shared semantics underlying natural language and programming language, and therefore undermines the trustworthiness of a model. In this paper, we first formally define the self-consistency of Code LLMs and then design a framework, IdentityChain, which effectively and efficiently evaluates the self-consistency and conventional accuracy of a model at the same time. We study eleven Code LLMs and show that they fail to preserve self-consistency, which is indeed a distinct aspect from conventional accuracy. Furthermore, we show that IdentityChain can be used as a model debugging tool to expose weaknesses of Code LLMs by demonstrating three major weaknesses that we identify in current models using IdentityChain. Our code is available at https://github.com/marcusm117/IdentityChain.

## 1 INTRODUCTION

Code Large Language Models (Code LLMs) are being increasingly employed in real-life applications (GitHub, 2023; OpenAI, 2023). Hence, evaluating them rigorously is a crucial problem. Conventional evaluations of Code LLMs focus on the models' accuracy on a wide range of individual tasks (Lu et al., 2021; Zhu et al., 2022), primarily the following two:

1) Code Generation *i.e.* Natural Language to Programming Language (NL-to-PL) Generation: Given a natural language specification, the model is tasked to generate a corresponding program.

2) Code Summarization *i.e.* Programming Language to Natural Language (PL-to-NL) Generation: Given a program, the model is tasked to generate a corresponding natural language specification.

However, evaluating these two tasks in isolation overlooks their symmetric nature. NL-to-PL and PL-to-NL Generation can be thought of as semantic-preserving translation and back-translation between the PL space and the NL space. Therefore, a trustworthy model should be able to correctly perform PL-to-NL Generation given programs generated by itself from previous NL-to-PL tasks. Similarly, it should correctly perform NL-to-PL Generation given natural language specifications generated by itself from previous PL-to-NL tasks. We call such a property "self-consistency".

Consider a real example shown in Figure 1. GPT-3.5 is first instructed to generate a program $pl_0$ according to a specification $nl_0$ written in a docstring, and then instructed to summarize its own code $pl_0$ into a new docstring $nl_1$. If we evaluate NL-to-PL and PL-to-NL Generation in isolation, GPT-3.5 is more than capable as it achieves $100\%$ accuracy on both tasks. However, from the self-consistency perspective, even though the model is self-consistent when generating $nl_1$ from $pl_0$,

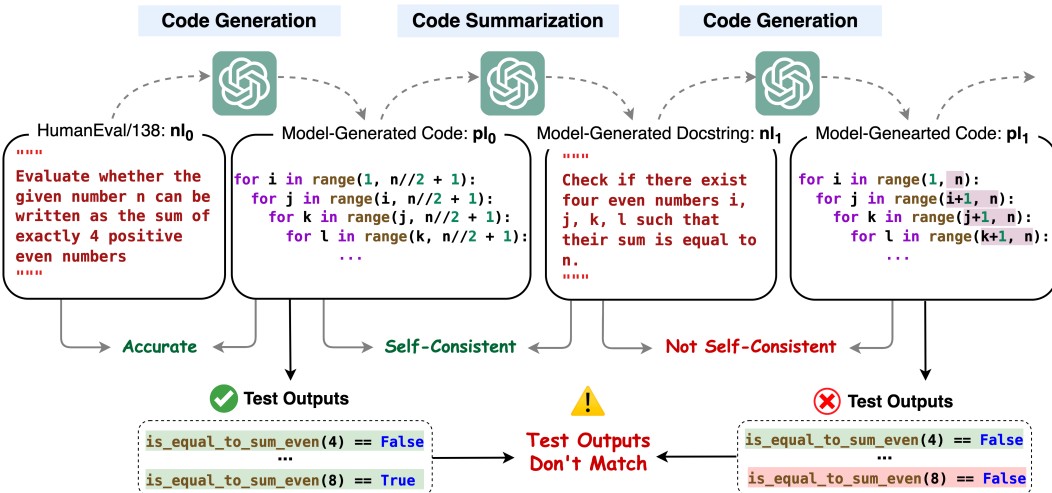

Figure 1: The IdentityChain Framework. Starting from a docstring $nl_0$, instruct the model to generate a program $pl_0$, summarize $pl_0$ into a new docstring $nl_1$, and generate a new program $pl_1$. If the test outputs of $pl_1$ do not match the ones of $pl_0$, then the model is not self-consistent. This chain can be extended to length $n \in \mathbb{N}$ and we compute whether, for all $i < n$, the test outputs of $pl_i$ match the ones of $pl_{i+1}$, returning a binary result that indicates if the model is self-consistent regarding $nl_0$.

it surprisingly fails to preserve self-consistency when generating $pl_1$ from its own docstring $nl_1$. Note that **self-consistency is different from consistency:** $nl_1$ here is generated by the model itself instead of arbitrarily crafted by humans or synthesized by other algorithms. This example reveals that GPT-3.5 doesn't understand the underlying semantics of the programs and natural language specifications, which raises a significant trustworthiness concern.

Unfortunately, current NL-to-PL evaluations (Chen et al., 2021; Li et al., 2023; Rozière et al., 2023) typically assess if the model-generated programs pass a set of test cases, and current PL-to-NL evaluations (Ahmad et al., 2021; Li et al., 2023; Rozière et al., 2023) commonly employ token-based metrics like BLEU (Papineni et al., 2002), which both fail to take self-consistency into account. Although similar self-consistency properties of LLMs have been probed through some natural language tasks (Jiang et al., 2023; Ohmer et al., 2023), their evaluations rely on Closed-domain QA tasks and cannot be generalized to open-ended generation (Section 2). Therefore, in this paper:

1) We formalize the definition of self-consistency and its evaluation (Section 3).

2) We design a novel framework, IdentityChain (Section 4), which effectively and efficiently evaluates a Code LLM's self-consistency by employing a new metric, Test Output Match (TOM) score, and leveraging greedy decoding during inference. Through experiments, we exhibit the effectiveness of the TOM score (Section 6.2) and the efficiency of greedy decoding (Section 6.3).

3) We evaluate eleven current Code LLMs including GPT-4, showing that they are not always self-consistent. Furthermore, we find that more accurate models are not necessarily more self-consistent, highlighting that self-consistency is a different aspect from conventional accuracy (Section 6.1).

4) We show through experiments that TOM score is also an effective metric to evaluate PL-to-NL Generation (Section 6.2), thus completing IdentityChain as a holistic framework that evaluates the NL-to-PL accuracy, PL-to-NL accuracy, and self-consistency of Code LLMs at the same time. We further discuss three major weaknesses of current models that we identify using IdentityChain, demonstrating the potential of IdentityChain as a debugging tool that helps model developers by exposing weaknesses of models and inspiring potential improvements (Section 6.4).

## 2 RELATED WORK

**Evaluating Code Large Language Models.** For NL-to-PL evaluation, token-based metrics like Exact Match (Ding et al., 2023b), Edit Distance (Zhang et al., 2023), Jaccard Similarity (Pei et al., 2023), and BLEU (Iyer et al., 2018; Ahmad et al., 2021) are used, but these metrics fail to capture the code-specific characteristics. To address this issue, CodeBLEU (Ren et al., 2020) takes Key-

words, Abstract Syntax Tree, and Data-Flow Match into account, and CodeBERTScore (Zhou et al., 2023) computes a similarity score of code embeddings extracted by pre-trained Code LLMs. However, static code similarity doesn't reflect the dynamic semantics of programs, which gave rise to execution-based metrics like Pass@K (Chen et al., 2021; Austin et al., 2021; Hendrycks et al., 2021; Li et al., 2022). Nonetheless, all existing NL-to-PL metrics focus only on the one-time accuracy while overlooking whether they are self-consistent regarding a model's own output. For PL-to-NL evaluation, BLEU (Papineni et al., 2002) score has been the automated metric adopted by most models (Rozière et al., 2023; Li et al., 2023; Wang et al., 2023). Metrics like ROGUE (Lin, 2004), chrF (Popović, 2015), and BERTScore (Zhang et al., 2020) are also reasonable choices. However, these static metrics fail to capture semantics separately from syntax and require ground truth references for comparison. In this paper, we proposed a dynamic metric TOM score for self-consistency evaluation, showing that it is not only effective but also compatible with all existing evaluation benchmarks with test cases. We also show that TOM score effectively evaluates PL-to-NL Generation regardless of ground-truth references, outperforming all aforementioned PL-to-NL metrics.

**Evaluating Self-Consistency of Large Language Models.** Previous studies (Minervini & Riedel, 2018; Li et al., 2019; Asai & Hajishirzi, 2020) show that LLMs behave inconsistently when given two semantically-bonded inputs.[1] However, measuring those inconsistencies is different from evaluating a model's self-consistency since these inputs, either hand-crafted or algorithm-synthesized are not generated by the model itself. As LLMs become better at multitasking (Brown et al., 2020; Ouyang et al., 2022), their self-consistency across tasks evolves into an important issue. Jiang et al. (2023) asks LLMs to generate the answer for an arithmetic reasoning problem, replace a variable in the original problem with an unknown $x$, and then instruct the same model to solve for $x$ given the answer it previously generated. Ohmer et al. (2023) asks LLMs to translate a question from English to another language and instruct the same model to answer the questions in both languages. However, both evaluation settings above rely on tasks with fixed ground truths and cannot be generalized to open-ended generation tasks where there can be multiple ground truth answers with arbitrary lengths. In this paper, we evaluate Code LLMs on two major open-ended generation tasks: NL-to-PL and PL-to-NL Generation.

## 3 FORMALIZATION

### 3.1 SELF-CONSISTENCY DEFINITION

Given a model $M$ that is capable of performing both NL-to-PL and PL-to-NL Generation, let $n2p$ and $p2n$ denote two instructions that respectively set $M$ to perform NL-to-PL Generation or PL-to-NL Generation. In practice, the instructions $n2p$ and $p2n$ are usually prompts. Therefore, a model instructed to perform one of the two tasks can be defined as two functions:

$$M_{n2p} : \mathcal{NL} \rightarrow \mathcal{PL} \qquad M_{p2n} : \mathcal{PL} \rightarrow \mathcal{NL}$$

where $\mathcal{PL}$ denotes the space of all valid programs in a specific programming language and $\mathcal{NL}$ denotes the space of all semantically valid and unambiguous[2] program specifications in a specific natural language. For example, $\mathcal{PL}$ can be the space of all valid Python programs and $\mathcal{NL}$ can be the space of all valid and unambiguous corresponding English specifications of these programs, which is the setting for all experiments later in this paper.

Let $nl_0 \in \mathcal{NL}$ be a valid and unambiguous natural language specification, and $pl_0 = M_{n2p}(nl_0)$ be the program generated by the model $M$ for $nl_0$. If the model is accurate, then $pl_0$ and $nl_0$ should have the same underlying semantics.[3] If we further instruct the model to generate a specification $nl_1 = M_{p2n}(pl_0)$ given $pl_0$, then the semantics of $pl_1, nl_1, pl_0$ should be all identical. We call such a property "self-consistency". Generally, a self-consistent model should be able to perform such translations between $\mathcal{NL}$ and $\mathcal{PL}$ infinitely many times without changing underlying semantics.

Note that self-consistency is a different property from accuracy. While accuracy assesses a model's ability to uni-directionally translate from $\mathcal{NL}$ to $\mathcal{PL}$ or the converse in a single step, self-consistency

---

[1] One input entails, contradicts, or is identical to the other.

[2] Nonsensical and ambiguous text is important in natural languages, but for NL-PL tasks, it makes more sense to only consider a subset of the natural language that validly and unambiguously specifies programs.

[3] Aside from program semantics *i.e.* input-output behavior, $nl_0$ and $pl_0$ should be also aligned regarding pragmatic aspects like complexity, security, and human readability. In this paper, our scope is just the semantics.

assesses the model's ability to bidirectionally translate between the two spaces in infinitely many steps. Therefore, a model can remain self-consistent even when it's inaccurate, as long as it consistently preserves the same error. Similarly, low self-consistency but high accuracy can also happen.

We can now formalize the above intuitions about the self-consistency of Code LLMs. Assume that given $\mathcal{NL}$ and $\mathcal{PL}$, there exists a semantics space $\mathcal{D}$ (we don't assume any specific definition of $\mathcal{D}$) *s.t.* an interpretation function $sem$ is well-defined as the following:

$$sem : \mathcal{NL} \cup \mathcal{PL} \to \mathcal{D}$$

which means that for all $pl \in \mathcal{PL}$ or $nl \in \mathcal{NL}$, the interpretation function $sem$ maps it uniquely to an element in $\mathcal{D}$. We define the self-consistency property as the following:

**Definition 1: Self-Consistency.** Given a valid and unambiguous specification $nl_0 \in \mathcal{NL}$, a model $M$ is self-consistent *w.r.t.* $nl_0$ if and only if

$$\forall i \in \mathbb{N}, \ sem(pl_i) = sem(nl_{i+1}) = sem(pl_{i+1})$$

where

$$pl_0 = M_{n2p}(nl_0), \quad nl_{i+1} = M_{p2n}(pl_i), \quad pl_{i+1} = M_{n2p}(nl_{i+1})$$

Aligning with the informal intuitions, our definition doesn't consider the initial generation $pl_0$ to be semantically the same as $nl_0$. As long as, for all $i \in \mathbb{N}$, the three-tuple $pl_i$, $nl_{i+1}$, and $pl_{i+1}$ are semantically identical, we can say that $M$ is self-consistent *w.r.t.* $nl_0$. If $pl_0$ is semantically identical to $nl_0$ and the model is self-consistent *w.r.t.* $nl_0$, we can say the model is "strong self-consistent", since if a model is always accurate, it must be self-consistent. We formally define it as:

**Definition 2: Strong Self-Consistency.** Given $nl_0 \in \mathcal{NL}$, a model $M$ is strong self-consistent *w.r.t.* $nl_0$ if and only if $M$ is self-consistent *w.r.t.* $nl_0$ and $sem(nl_0) = sem(pl_0)$, where $pl_0 = M_{n2p}(nl_0)$.

Similar to the above definitions, we can further define self-consistency and strong self-consistency *w.r.t.* an arbitrary $pl_0 \in \mathcal{PL}$. Note that these two classes of definitions are not equivalent,[4] but for simplicity, we adopt the self-consistency and strong self-consistency *w.r.t.* $nl_0 \in \mathcal{NL}$ definitions.

## 3.2 Self-Consistency Evaluation

**Chain of Identity Transformations.** Let a model $M$ be self-consistent *w.r.t.* $nl_0$. Instruct the model to generate $pl_0 = M_{n2p}(nl_0)$ and iteratively apply the PL-to-NL function to get $nl_{i+1} = M_{p2n}(pl_i)$ and the NL-to-PL function to get $pl_{i+1} = M_{n2p}(nl_{i+1})$. From the semantics perspective, alternatively applying the PL-to-NL and NL-to-PL functions on $pl_0$ for $n \in \mathbb{N}^+$ times is equivalent to applying the identity transformation $I$ in the semantics space $\mathcal{D}$ on $sem(pl_0)$ for $2n$ times:

$$sem((M_{n2p} \circ M_{p2n})^n(pl_0)) = I^{2n}(sem(pl_0))$$

The chain of transformations on $pl_0$ between the language spaces $\mathcal{NL}$ and $\mathcal{PL}$ corresponds to a chain of identity transformations on $sem(pl_0)$ within the semantics space $\mathcal{D}$. In the equation above, the superscript $n$ denotes the length of such an "identity chain".

**Self-Consistency Scores.** To evaluate the self-consistency of a model $M$, it's impossible to extend the identity chain infinitely long or exhaust all $\mathcal{NL}$, so we approximate by picking a fixed chain length $n \in \mathbb{N}^+$ and a reasonably large subset of $\mathcal{NL}$ with $m \in \mathbb{N}^+$ elements as an evaluation set. We index the inputs in the evaluation set by $j \in \mathbb{N}^+$, $1 \leq j \leq m$. For an input $nl_{0,j}$ in the evaluation set, we check its corresponding semantic equalities $sem(pl_i) = sem(nl_{i+1}) = sem(pl_{i+1})$ for all $i \in \mathbb{N}, 0 \leq i < n$. We use a binary output $\text{sc}_{n,j} \in \{0, 1\}$ to indicate whether all semantic equalities are true at the same time *i.e.* whether $M$ is self-consistent *w.r.t.* $nl_{0,j}$ within $n$ steps. Similarly, we use $\text{ssc}_{n,j} \in \{0, 1\}$ to denote if $M$ is strong self-consistent *w.r.t.* $nl_{0,j}$ within $n$ steps. Finally, by aggregating $\text{sc}_{n,j}$ and $\text{ssc}_{n,j}$ over all $j$, we can evaluate self-consistency and strong self-consistency of $M$ within $n$ steps by reporting two scores $\text{SC}_n$ and $\text{SSC}_n$ defined as the following:

$$\text{SC}_n = \frac{\sum_{j=1}^{m} \text{sc}_{n,j}}{m} \qquad \text{SSC}_n = \frac{\sum_{j=1}^{m} \text{ssc}_{n,j}}{m}$$

---

[4]Self-consistency *w.r.t.* all $nl_0 \in \mathcal{NL}$ doesn't imply self-consistency *w.r.t.* all $pl_0 \in \mathcal{PL}$. The converse is also not true. The NL-to-PL function $M_{n2p}$ can simply map all $nl_0$ to the exact same $pl_0$, where $M$ is strong self-consistent *w.r.t.* $pl_0$. No claim can be made about $M$'s self-consistency *w.r.t.* the entire $\mathcal{PL}$ space.

# 4 THE IDENTITYCHAIN FRAMEWORK

## 4.1 EFFECTIVE SELF-CONSISTENCY EVALUATION

Determining the truth value of the semantic equalities $sem(pl_i) = sem(nl_{i+1}) = sem(pl_{i+1})$ can be performed by humans. However, it's not feasible to employ human judgment when the evaluation set scales up. Consequently, we need automated metrics as approximations.

**Inapplicability of Existing Automated Metrics.** Ideal automated PL-to-NL and NL-to-PL metrics should map a program and a natural language specification to the semantic space, and directly compute their semantic distance. Given such ideal metrics, we can approximate or even determine the truth values of $sem(pl_i) = sem(nl_{i+1})$ and $sem(nl_{i+1}) = sem(pl_{i+1})$. However, all existing metrics gauge the semantic equalities indirectly by computing a distance between the model-generated candidate and a predefined ground truth reference. Specifically, all existing NL-to-PL metrics compute a distance between two programs in the same programming language and all existing PL-to-NL metrics compute a distance between two specifications in the same natural language. Unfortunately, we do not have any predefined ground truth reference for either $nl_{i+1}$ or $pl_{i+1}$.[5]

**Relaxation of the Semantic Equalities.** Recall that our goal is to approximate the truth value of the semantic equalities $sem(pl_i) = sem(nl_{i+1}) = sem(pl_{i+1})$. Although there are no existing metrics to approximate the truth values of $sem(pl_i) = sem(nl_{i+1})$ or $sem(nl_{i+1}) = sem(pl_{i+1})$, the third equality $sem(pl_i) = sem(pl_{i+1})$ is feasible to gauge. We can use existing NL-to-PL metrics to approximate this equality as they directly compute a distance between two programs in the same programming language. In addition, if the model summarizes $pl_i$ wrongly into a semantically different $nl_{i+1}$, then program $pl_{i+1}$, which is supposed to be semantically identical to $nl_{i+1}$, is highly unlikely to have the exact same semantics as $pl_i$ and vice versa. Therefore, any effective NL-to-PL metric, which approximates the truth value of $sem(pl_i) = sem(pl_{i+1})$, can be also considered as an effective approximation to that of $sem(pl_i) = sem(nl_{i+1})$. In Table 2, we empirically show that there is a positive correlation between them.

**Design of the Test Output Match (TOM) Score.** While all NL-to-PL metrics have the potential to be self-consistency evaluation metrics, we want to pick one that best approximates the semantic equality $sem(pl_i) = sem(pl_{i+1})$. As reviewed in Section 2, execution-based dynamic metrics like Pass/Fail can directly, though not complete, gauge the code semantics, and are therefore more preferred than static metrics like CodeBLEU. In Table 2, we empirically verify this conclusion.

The most widely used dynamic metric, Pass@K, is not directly applicable to self-consistency evaluation. Whether $pl_i$ passes or fails the test cases does not imply whether it is semantically identical to $pl_{i+1}$ and vice versa, so naturally, we come up with a new metric, the Pass/Fail Match (P/FM) score, which checks if $pl_i$ and $pl_{i+1}$ both pass or both fail at the same time. If both of them pass all test cases, they must be semantically identical. If one passes while the other fails, they must be semantically different. However, P/FM doesn't handle the Fail-Fail situation well since $pl_i$ and $pl_{i+1}$ can fail the same test case due to completely different reasons.

We, therefore, propose another new metric, the Test Output Match (TOM) score, which compares the exact output of $pl_i$ and $pl_{i+1}$ for each test case, records 1 if the outputs match and 0 if the outputs differ, and finally computes the percentage of matches among all test cases.

$$\text{TOM} = \frac{\text{Number of Matched Outputs}}{\text{Total Number of Test Cases}}$$

For syntax errors and runtime errors like ValueError or IndexError, the TOM score is calculated by comparing the full error message instead of just the error type. By capturing more fine-granular semantic information, TOM score better approximates the truth value of $sem(pl_i) = sem(pl_{i+1})$ than the simple P/FM score. In Table 2, we show that TOM indeed better correlates to the human-judged truth value, and therefore is an effective metric for self-consistency evaluation.

---

[5]Taking $nl_i$ or $pl_i$ as the ground truth reference for $nl_{i+1}$ or $pl_{i+1}$ is not generally applicable. For example, if $pl_0$ fails some test cases, then $nl_1 = M_{p2n}(pl_0)$, which is supposed to be semantically identical to $pl_0$, must be semantically different from $nl_0$. Therefore, $nl_0$ cannot be seen as the ground truth for $nl_1$.

## 4.2 EFFICIENT SELF-CONSISTENCY EVALUATION

**Efficient Evaluation by Greedy Decoding.** To evaluate self-consistency up to a certain chain length $n$, we use greedy decoding for both NL-to-PL and PL-to-NL Generation. Given a starting point $nl_0$, if at some step $i$ in the chain, $pl_{i+1}$ is an exact match of $pl_i$, or $nl_{i+1}$ is an exact match of $nl_i$, then by the deterministic nature of greedy decoding, we know that the model will always generate the same program and specification repeatedly. In such cases, we can assert that the model is self-consistent *w.r.t.* $pl_i$ or $nl_i$ (not necessarily $nl_0$). Therefore, our IdentityChain framework adopts greedy decoding and stops the chain early when exact matches are found. We show in Figure 2 that, with greedy decoding and early stopping, self-consistent cases can be quickly determined.

## 4.3 HOLISTIC EVALUATION OF CODE LLMS

The IdentityChain framework not only effectively and efficiently evaluates the self-consistency of a Code LLM, but also holistically evaluates multiple aspects of a model at the same time.

**NL-2-PL Accuracy.** The bootstrapping step from $nl_0$ to $pl_0$ is exactly the canonical NL-to-PL evaluation setting, where we can compute the Pass@1 score to evaluate the model's NL-to-PL accuracy.

**PL-2-NL Accuracy.** Unlike NL-to-PL metrics, existing PL-to-NL metrics are all static and therefore struggle to capture underlying semantics. As discussed in Section 4.1, by back-translating a model-generated natural language specification into another program, we can approximate the semantic equality between the original program and the specification. Therefore, the $SC_1$ score *i.e.* the averaged TOM score between all $pl_0$ and $pl_1$, can be an effective metric for the model's PL-to-NL accuracy. In Table 2, we empirically show that TOM outperforms all existing PL-2-NL metrics.

**Strong Self-Consistency.** An ideal model should be both accurate and self-consistent. An accurate but not self-consistent model is not trustworthy, while a self-consistent but not accurate model is useless. The strong self-consistency score $SSC_n$ takes both accuracy and self-consistency into account, which serves as a comprehensive evaluation of the model's overall performance.

Model developers can first check the $SSC_n$ score as a performance summary and then examine the $SC_n$, Pass@1, and $SC_1$ scores to determine whether the model is lacking more accuracy or self-consistency. More importantly, with IdentityChain, it's easy to pinpoint cases where a model is not self-consistent to reveal subtle weaknesses of the model, as we will show in Section 6.4.

## 5 EXPERIMENTS

**Benchmarks.** We evaluate the self-consistency of Code LLMs on two widely adopted benchmarks: HumanEval and MBPP. HumanEval (Chen et al., 2021) contains 164 hand-crafted Python problems. Liu et al. (2023) proposes HumanEvalPlus to augment HumanEval with more test coverage. Specifically, we use HumanEvalPlus-Mini-v0.1.6 where each problem has 16.5 test cases on average. MBPP Austin et al. (2021) includes 974 crowd-sourced Python problems with 3.0 test cases for each problem on average. For more precise evaluations, we use the test split of the sanitized version of MBPP, which contains 257 problems manually verified by Austin et al. (2021). In both datasets, all problems have predefined meaningful function names, for example, "has_close_elements". If the model generates an incorrect function body at the initial step, there can be a conflict between the semantics of the function body and the name, which weakens the soundness of self-consistency evaluation. Therefore, we replace meaningful function names with a generic "func" at all steps except the initial one, so that the model solely relies on the semantics of the function body or docstring instead of taking shortcuts using the function name. See Appendix C for a concrete example.

**Models.** We evaluate two types of Code LLMs: foundation models and instruction-tuned models. For foundation models, we evaluate two open-source model families, StarCoderBase (Li et al., 2023) and Code Llama (Rozière et al., 2023). For instruction-tuned models, we evaluate the instruction-tuned versions of Code Llama and StarCoderBase, Google's Gemini-1.0-Pro-001[6] (Team, 2023), and three most capable OpenAI models: GPT-3.5-Turbo-0613, GPT-4-0613, and GPT-4-0125-

---

[6]We set the temperature to 0.2 for Gemini since the API sometimes returns no response due to its recitation or safety filtering mechanism. To compare with other models using temperature 0.2, see Figure 3 and 5.

Preview (the latest GPT-4-Turbo snapshot). For models from Google and OpenAI, we choose the parameter-frozen snapshots of them so that the results can be reproduced.

**Prompts.** We use one-shot prompting for all the models on both benchmarks to better guide the model to generate the expected format.[7] For instruction-tuned models, we formulate the prompt as chats (Ouyang et al., 2022), where the "system" role provides general instructions, the "user" role provides the input of the one-shot example, and the "assistant" role provides the output of the one-shot example. For foundation models, the prompt is only the one-shot example. To maximize the capacity of all Code LLMs, we carefully customize the prompt template for each model. See the "examples" folder in our code repository for details of the prompt templates. See Appendix B for detailed hardware and software configurations of all experiments.

# 6 RESULTS

## 6.1 SELF-CONSISTENCY OF CODE LLMS

**Code LLMs Fail to Preserve Self-Consistency.** We observe in Table 1 that all models' self-consistency and strong self-consistency decreases as the number of iteration steps increases. For example, all models' $SSC_5$ scores, which assess strong self-consistency within five steps, evidently decline up to 78.0% compared to the initial Pass@1.[8] Regardless of the accuracy of the initial generation, all models' $SC_5$ scores, which assess self-consistency within five steps, also decline up to 43.8% compared to $SC_1$. Such a performance drop indicates that while the models might be initially (strong) self-consistent, they are not able to preserve it. In Section 6.4, we delve deeper into the some of root-cause errors that trigger violations of (strong) self-consistency.

| Model | Size | HumanEvalPlus | | | | MBPP Sanitized | | | |
|---|---|---|---|---|---|---|---|---|---|
| | | Pass@1 | $SSC_5$ | $SC_1$ | $SC_5$ | Pass@1 | $SSC_5$ | $SC_1$ | $SC_5$ |
| | | | | | Instruction-tuned Models | | | | |
| **Gemini-Pro**[6] | N/A | 54.6 | 20.2 ↓63.0% | 44.2 | 27.6 ↓37.6% | 57.2 | 30.7 ↓46.3% | 61.9 | 44.4 ↓28.3% |
| **GPT-4-Turbo** | N/A | 81.0 | 59.5 ↓26.5% | 81.0 | 68.1 ↓15.9% | 73.9 | 61.5 ↓16.8% | 85.6 | 77.8 ↓9.1% |
| **GPT-4** | N/A | 74.8 | 63.8 ↓14.8% | 84.0 | 76.1 ↓9.5% | 72.8 | 62.6 ↓13.9% | 88.7 | 82.5 ↓7.0% |
| **GPT-3.5** | N/A | 71.8 | 40.5 ↓43.6% | 56.4 | 50.3 ↓10.9% | 68.9 | 54.9 ↓20.3% | 86.4 | 76.3 ↓11.7% |
| **CodeLlama-Inst** | 7B | 16.0[8] | 4.3 ↓73.1% | 17.8 | 14.1 ↓20.7% | 22.2 | 11.7 ↓47.4% | 30.7 | 25.3 ↓17.7% |
| | 13B | 30.7 | 17.8 ↓42.0% | 40.5 | 33.1 ↓18.2% | 40.5 | 23.0 ↓43.3% | 50.2 | 42.8 ↓14.7% |
| **StarChat-Beta** | 15B | 25.2 | 5.5 ↓78.0% | 19.6 | 11.0 ↓43.8% | 32.3 | 7.8 ↓75.9% | 14.8 | 11.3 ↓23.7% |
| | | | | | Foundation Models | | | | |
| **CodeLlama** | 7B | 23.9[8] | 8.0 ↓66.7% | 22.1 | 19.0 ↓13.9% | 38.9 | 20.6 ↓47.0% | 45.1 | 43.6 ↓3.4% |
| | 13B | 35.6 | 9.8 ↓72.4% | 17.8 | 14.1 ↓20.7% | 46.3 | 23.0 ↓50.4% | 47.9 | 42.0 ↓12.2% |
| **StarCoderBase** | 1B | 11.0 | 3.7 ↓66.7% | 12.3 | 9.8 ↓20.0% | 28.8 | 11.3 ↓60.8% | 34.2 | 31.5 ↓8.0% |
| | 3B | 17.8 | 4.9 ↓72.4% | 12.3 | 11.0 ↓10.0% | 37.4 | 14.4 ↓61.5% | 39.3 | 34.2 ↓12.9% |
| | 7B | 24.5 | 8.6 ↓65.0% | 19.0 | 16.0 ↓16.1% | 43.6 | 23.0 ↓47.3% | 47.1 | 43.6 ↓7.4% |
| | 15B | 27.0 | 8.0 ↓70.5% | 20.9 | 17.2 ↓17.6% | 44.0 | 21.0 ↓52.2% | 44.7 | 41.2 ↓7.8% |

Table 1: Performance of Code LLMs evaluated by IdentityChain. Pass@1 indicates the NL-to-PL accuracy. $SC_1$ representing self-consistency within 1 step indicates PL-to-NL accuracy. $SC_5$ represents self-consistency within 5 steps and $SSC_5$ represents strong self-consistency within 5 steps.

**Self-Consistency is Different from Conventional Accuracy.** Existing evaluations of Code LLMs refer to conventional accuracy (*e.g.* Pass@K) as the model's overall capacity, which is confirmed by our results in Table 1: larger models in the same model families indeed have higher Pass@1 scores. However, results in Table 1 show that stacking more parameters does not necessarily guarantee improvement of self-consistency. For example, the Pass@1 score of StarChat-Beta (15B), which indicates accuracy, is higher than Code Llama-Instruct-7B for both benchmarks, but the $SC_5$ score of the former, which indicates self-consistency, is lower than the latter for both benchmarks. For another example, while StarCoderBase-7B performs worse than StarCoderBase-15B in Pass@1 for both benchmarks, it outperforms the double-sized version of itself in $SSC_5$, which indicates strong self-consistency, for both benchmarks.

---

[7]For MBPP, we use task 2 in the prompt split as the one-shot example. For HumanEvalPlus, since there's no dedicated prompt split, we use HumanEval/0 as the one-shot example and exclude it from experiments.

[8]For Code Llama-Instruct and Code Llama 7B, the Pass@1 we measured are noticeably different from those reported by Rozière et al. (2023). We conjecture that it might be caused by the models' sensitivity to prompts.

Moreover, conventional accuracy can underestimate the capability difference between models, and self-consistency complements the drawback. For example, GPT-4, which is recognized to be significantly more capable than GPT-3.5, reports a Pass@1 score of 74.8 on HumanEvalPlus, which is only a 4.2% relative improvement compared to GPT-3.5. However, GPT-4 is significantly more self-consistent. It achieves an $SC_5$ score of 76.1, which is 51.2% higher than GPT-3.5, highlighting that there is a non-trivial capability gap between GPT-4 and GPT-3.5.

## 6.2 EFFECTIVENESS OF TOM SCORE

To show the effectiveness of TOM, we excerpt a 1-step chain $(nl_0, pl_0, nl_1, pl_1)$ from an Identity-Chain experiment of GPT-3.5, and gathered human-judged ground truth of whether $nl_1$ is semantically identical to $pl_0$ *i.e.* $sem(pl_0) = sem(nl_1)$.[9]

| Metric | $r$ | $\rho$ | $\tau$ |
|---|---|---|---|
| EM | .285 | .285 | .285 |
| CodeBLEU | .179 | .173 | .144 |
| P/FM | .294 | .292 | .292 |
| TOM | **.461** | **.454** | **.410** |

| Metric | $r$ | $\rho$ | $\tau$ |
|---|---|---|---|
| BLEU | .285 | .275 | .227 |
| ROUGE-L | .254 | .239 | .196 |
| chrF | .271 | .256 | .210 |
| BERTScore | .381 | .389 | .319 |
| TOM | **.500** | **.482** | **.445** |

Table 2: Pearson ($r$), Spearman ($\rho$), and Kendall-Tau ($\tau$) correlations with human-judged ground truth of whether $pl_0$ is semantically identical to $nl_1$, the model-generated docstring for $pl_0$.

**TOM is Effective for Self-Consistency Evaluation.** We compared TOM to two static PL space metrics: Exact Match (EM) and CodeBLEU and the naive dynamic metric Pass/Fail Match (P/FM) using Pearson ($r$), Spearman ($\rho$), and Kendall-Tau ($\tau$) correlations with human judgment in Table 2. Recall our conjectures in Section 4.1: PL space metrics can approximate the truth value of $sem(pl_0) = sem(nl_1)$, dynamic metrics are better than static ones, and the fine-grained TOM score is better than naive P/FM. All three conjectures are verified in this experiment.

**TOM is Effective for PL-to-NL Evaluation.** Within the same experiment setting, we compare TOM with four NL space metrics: BLEU, ROUGE-L, chrF, and BERTScore in Table 2. Note that for this comparison, the correlations are only computed on 117 out of the total 163 problems where $pl_0$ passes all test cases. Otherwise, if $pl_0$ is not semantically the same as $nl_0$, we can't use $nl_0$ as a ground truth reference for $nl_1$ to calculate those NL space metrics. We show that TOM outperforms all NL space metrics given that their ground truth references exist, not to mention that TOM works well for the remaining 46 problems, for which the ground truth references are absent.

## 6.3 EFFICIENCY OF GREEDY DECODING

**Greedy Decoding Efficiently Evaluates Self-Consistency.** We find that using greedy decoding, IdentityChain efficiently reveals most not self-consistent cases within the initial three steps. Figure 2 shows an evident decline of both $SC_i$ and $SSC_i$ scores within the first three steps. After that, all models stabilize or show only minimal decreases in their (strong) self-consistency scores, which underscores the efficiency of IdentityChain as an evaluation tool. Although we set the chain length to five in our experiments, for mode developers and researchers with tighter time limits or computing resources, it's reasonable to choose a shorter chain length when using greedy decoding.

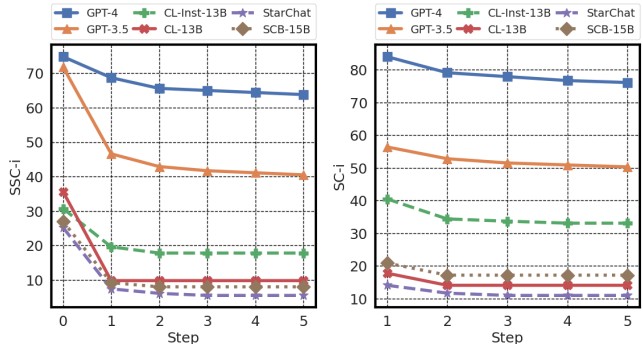

Figure 2: $SSC_i$ and $SC_i$ at Computed Each Step $i$.

---

[9]Different from the grading policy used in Chen et al. (2021), which ignores incorrect input-output examples in the generated docstrings, we consider a docstring with correct description but wrong examples still wrong.

**Greedy Decoding Results are Generalizable to Different Temperatures.** To show the generalizability of greedy decoding results, we additionally evaluate the $SC_5$ score at four different temperatures. As illustrated in Figure 3, while the $SC_5$ scores of all models decrease as the temperature increases, their relative rankings mostly remain *i.e.* more self-consistent models are always more self-consistent regardless

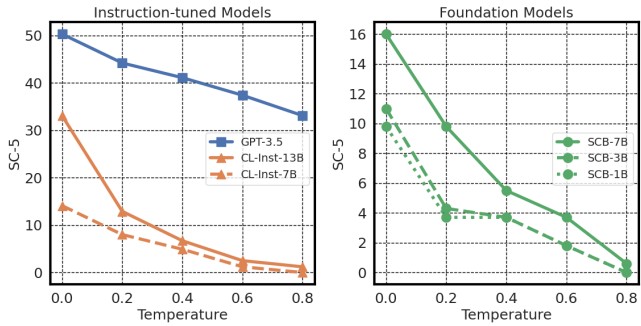

Figure 3: $SC_5$ Evaluated at Different Temperatures.

of temperature, which shows that the greedy decoding results are indeed generalizable. Moreover, it is reasonable that the absolute self-consistency of all models drops as the temperature increases. There is always a balance between exploration, which introduces novel solutions but weakens self-consistency, and exploitation, which ensures self-consistency but may overlook novel solutions.

From our observations, greedy decoding is both more efficient and more appropriate for self-consistency evaluation. See Appendix D for the $SSC_5$ scores evaluated at different temperatures.

## 6.4    IDENTITYCHAIN AS A MODEL DEBUGGING TOOL

Evaluating Code LLMs with IdentityChain, we can easily pinpoint the cases where the model is not self-consistent. Studying these non-self-consistent cases, we identify three major weaknesses of current models in code understanding, which are not captured by accuracy-oriented evaluations.

**Code LLMs Have Weak Sense of Data Types.** We observe that Code LLMs are not sensitive to data types. In all programming languages, data type is a fundamental element that specifies how variables should be stored, manipulated, and interacted with. However, we find that current models tend to overlook data type information. We show such an example in Appendix Figure 6. Inaccurate interpretations of data types will inevitably result in erroneous code usage. In real software development scenarios, it can lead to intricate issues like memory management, performance bottlenecks, or unexpected behaviors during code execution  (Ding et al., 2022; He & Vechev, 2023).

**Code LLMs Have Weak Sense of Implicit Code Semantics.** We observe that Code LLMs cannot accurately capture the implicit code semantics, which is a major root cause of non-self-consistency. Current models tend to only capture the shallow semantics that are explicitly presented in the program while overlooking the implicit logic. For example, they tend to only summarize the explicit if-checks while ignoring the implicit else-branch. We show two concrete examples in Appendix Figure 7. Ignoring implicit code semantics during PL-to-NL Generation will unavoidably result in misleading or ambiguous documentation in real development scenarios.

**Code LLMs Have Weak Sense of Code Execution.** We also observe that Code LLMs cannot accurately predict the execution outcomes Austin et al. (2021); Ding et al. (2023a). Specifically, when instructed to summarize programs, the models often generate correct natural language specifications but incorrect input-output examples. We show two concrete examples in Appendix Figure 8. This weakness is particularly concerning if we want to generate test cases to guide the entire software development process (Test-Driven Development), which underscores the importance of aligning the models' PL-to-NL Generation ability with their understanding of code execution.

## 7    CONCLUSION

In conclusion, we reveal that different from accuracy, self-consistency is indeed a crucial missing link in current evaluations of Code LLMs, and IdentityChain effectively and efficiently bridges the gap. More importantly, IdentityChain can be used not only as a holistic evaluation tool but also as a model debugging tool that helps model developers study weaknesses in their models and thus potentially inspire future improvements. See Appendix A for future directions that potentially extend the scope of self-consistency evaluation or improve current Code LLMs using IdentityChain.

ACKNOWLEDGEMENT

We would like to thank Qianyu Gu, Nan Jiang, and Sophia Su for valuable discussions. This work was supported in part by an IBM Ph.D. Fellowship, DARPA/NIWC-Pacific N66001-21-C-4018, NSF CNS–1845995, CNS-2247370, CCF-2221943, CCF-2313055, CCF-1845893, and CCF-2107405. Any opinions, findings, conclusions, or recommendations expressed herein are those of the authors and do not necessarily reflect those of IBM, DARPA, or NSF.

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

## A  FUTURE WORK

**Introducing PL-to-PL and NL-to-NL Generation.**  It is natural to extend the self-consistency definitions on a large set of multiple programming and natural languages by introducing PL-to-PL Generation *i.e.* Code Translation and NL-to-NL Generation *i.e.* Machine Translation. In practice, IdentityChain can be improved to support more programming languages and natural languages.

**Studying Weaknesses of Code LLMs.**  Following the three examples in Section 6.4, future work can further identify and categorize more subtle weaknesses in Code LLMs. More importantly, we encourage future work to investigate the relationship between those weaknesses and the training data. It is possible that the weaknesses are barely addressed within current training paradigms.

**Fine-tuning Code LLMs for Better Self-Consistency.**  It is not yet clear how we can improve the self-consistency of Code LLMs. For a model with imbalanced NL-to-PL and PL-to-NL accuracy, fine-tuning the task that the model performs worse can possibly work. For a model with balanced accuracy, we might need to customize a fine-tuning dataset that contains input-output pairs generated by the model itself. Many fair hypotheses can be made following this line of reasoning. We encourage future work to raise more and test them accordingly.

## B  EXPERIMENT CONFIGURATIONS

For all models, we use greedy decoding for our main experiment in Section 6.1. The closed-source OpenAI models GPT-3.5 and GPT-4 are non-deterministic and there is no way to set them to perform greedy decoding using APIs. Therefore, we set the temperature to 0 to minimize the randomness.

For open-source models, all model checkpoints are downloaded using the Python library "transformers" from Hugging Face. Note that we downloaded Code Llama and Code Llama-Instruct, from `https://huggingface.co/codellama` instead of the link provided by Meta AI. We run open-source model experiments on NVIDIA RTX A6000 GPUs with CUDA 11.3, cuDNN8-devel, PyTorch 1.12.1, and Python 3.10.9. For efficiency, we set the max prompt length to be 1,024 tokens, the max generation length to be 512 tokens, and the inference precision to be FP16.

## C  REPLACING MEANINGFUL FUNCTION NAMES

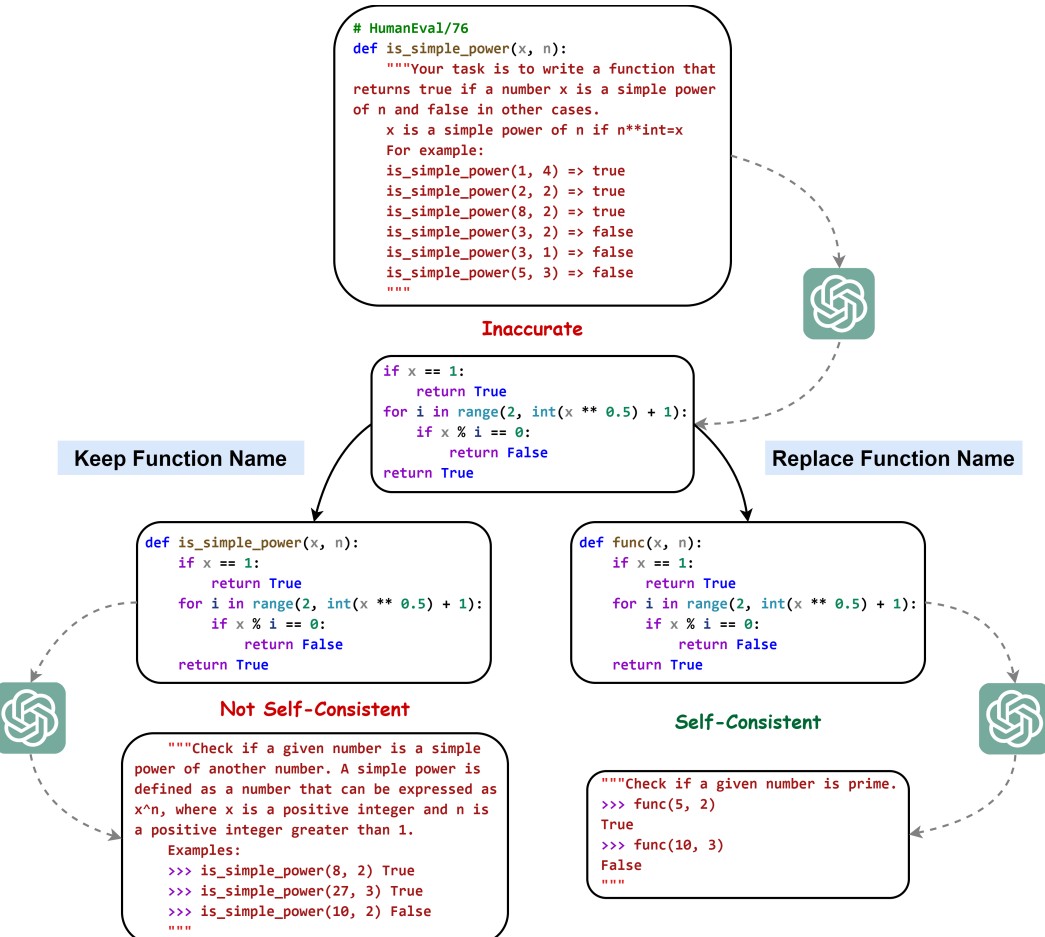

Figure 4: Replacing Meaningful Function Names with A Generic "func". Given the docstring with the original function name, GPT-3.5 generates an incorrect program that conflicts with the function name. When further summarizing that program along with the original function name, GPT-3.5 completely ignores the code and generates a new docstring based on the function name. In this case, we will falsely conclude that GPT-3.5 is not self-consistent. However, when summarizing the program along with a generic name "func" in replacement, GPT-3.5 correctly captures the code semantics and thus is self-consistent *w.r.t.* the original docstring. Therefore, when generating $nl_i$ and $pl_i$ for $i \geq 1$, we replace the original meaningful function name with the generic "func".

## D GENERALIZABILITY OF GREEDY DECODING

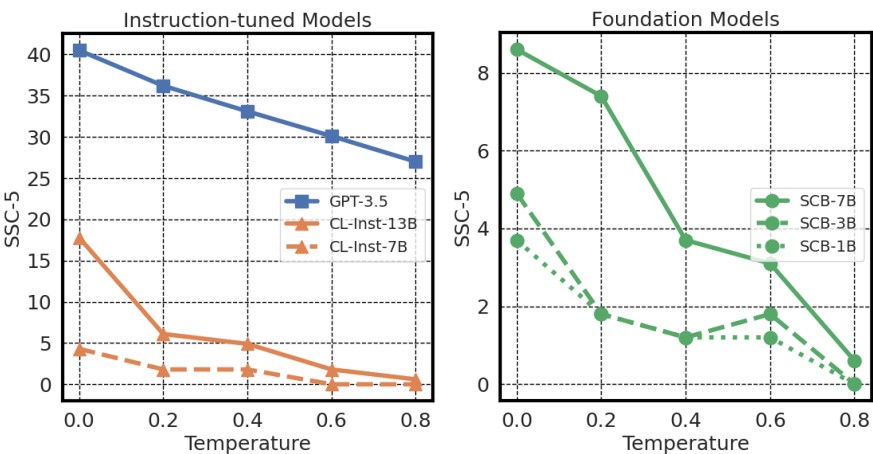

Figure 5: SSC$_5$ Evaluated at Different Temperatures. Similar to the SC$_5$ results in Section 6.3, for the strong self-consistency score SSC$_5$, the relative rankings of models mostly remain regardless of temperature *i.e.* more strong self-consistent models are always more strong self-consistent no matter the temperature, which confirms that greedy results are generalizable to different temperatures.

## E EXAMPLES OF CODE LLMS' WEAKNESSES

GPT-3.5 Implementation for HumanEval/92

```python
def func(x, y, z):
    if isinstance(x, int) and isinstance(y, int) and isinstance(z, int):
        if x == y + z or y == x + z or z == x + y:
            return True
        return False
```

GPT-3.5 Summarization of the Above Code

```python
"""Check if any two numbers in the given three numbers add up to the third number.
>>> func(1, 2, 3)
True
>>> func(4, 5, 6)
False
"""
```

**Fail to Capture The Data Type Checking**

Figure 6: Code LLMs Have Weak Sense of Data Types. The implementation checks whether all three inputs are type int at the same time, but the summarization only mentions that the inputs are three "numbers" failing to capture the data type information.

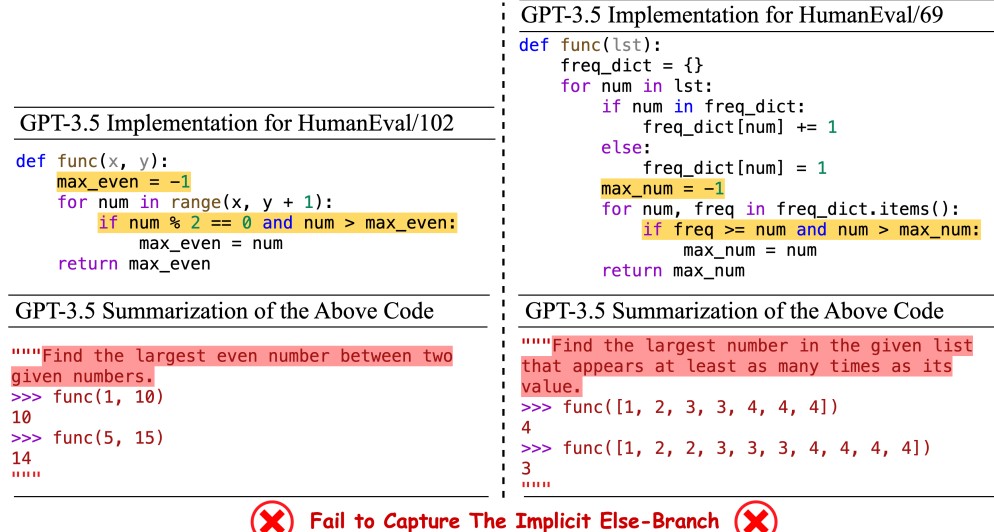

Figure 7: Code LLMs Have Weak Sense of Implicit Code Semantics. In the left example, the implementation has an implicit "else" branch that returns −1 when no even number is found. In the right example, the implementation also has an implicit "else" branch that returns −1 when no larger satisfying number is found. However, both summarizations fail to capture that implicit logic.

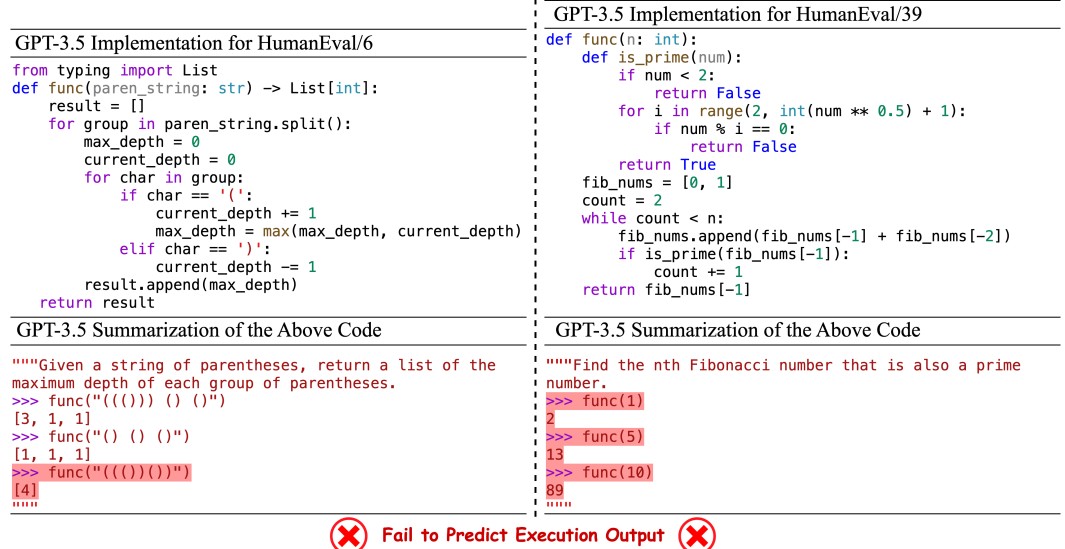

Figure 8: Code LLMs Have Weak Sense of Code Execution. In both examples, some input-output pairs in the summarization are wrong, which means that the model fails to predict execution.

