# OpenReview forum: "Beyond Accuracy: Evaluating Self-Consistency of Code Large Language Models with IdentityChain"
_ICLR.cc/2024/Conference — ICLR 2024 poster_

### Official Review · Reviewer_AByS · 2023-10-14

**Soundness:** 3 good
**Presentation:** 4 excellent
**Contribution:** 3 good
**Rating:** 8
**Confidence:** 4

**Summary:**

This paper presents a novel evaluation framework called IdentityChain which is designed mainly to evaluate self-consistency. Basically, given some initial natural language specification ($nl_0$), a model ($M$) will first generate code ($pl_0$). Then, the $M$ generates a natural language summary ($nl_1$), given only ($pl_0$). Now, using only $nl_1$, $M$ generates the corresponding code ($pl_1$). So, there is effectively a chain: $nl_0 \rightarrow pl_0 \rightarrow nl_1 \rightarrow pl_1$. Their framework measures semantic equivalence, where the self-consistency condition is met if $sem(pl_0) = sem(nl_1) = sem(pl_1)$, and the \textit{strong} self-consistency condition is met if $sem(nl_0) = sem(pl_0) = sem(nl_1) = sem(pl_1)$. This corresponds to a chain of length 1, but this could be extended to an arbitrary length (they use 5 in this paper). For measuring semantic equivalence $sem(pl_i) = sem(pl_{i+1})$, they introduce a metric called Test Output Match (TOM) in which they check whether the exact output of $pl_i$ and $pl_{i+1}$ are the same for each of the tests in the given test suite. They argue that this also implicitly evaluates $sem(pl_i) = sem(nl_{i+1})$. They conduct experiments using HumanEvalPlus and MBPP, using 11 different code LLMs, encompassing both foundational models as well as instruction-tuned models using greedy decoding. Their results show that even though code LLMs can achieve impressive initial accuracy (pass@1), they achieve relatively lower scores on self-consistency, especially as the number of iterations increase, and in fact, accuracy does not always correlate with self-consistency. They demonstrate that IdentityChain can be used for efficiently finding flaws at each step by highlighting a few of their findings from such experiments. They also demonstrate that self-consistency is better correlated with human judgment compared to other common static and dynamic metrics.

**Strengths:**

- This notion of consistency is very interesting and very useful for evaluation. The idea of using the Test Output Match for approximating semantic equivalence is also quite neat, and something particularly useful for evaluating PL-to-NL since existing metrics are not great for this.
- The analysis in Figure 2 for understanding how performance degrades as the number of iterations increases is quite interesting. The authors have also enumerated a number of findings related to weaknesses of code LLMs from their own use of this framework which can inspire future work in this space.
- The authors have conducted very thorough experiments and have in-depth evaluation, even demonstrating correlation with human judgment (though some of the details of this are unclear). This underlines the value of the evaluation framework they have introduced.

**Weaknesses:**

- The IdentityChain framework assumes that the underlying code model performs well at both tasks (NL-to-PL and PL-to-NL). It is possible that the model is good at one and not the other, and it is not possible to decouple these in their framework.
- The authors introduce the idea in a very general manner, in which the self-consistency idea could be applied between many different tasks. However, it is important to note that this approach can only be applied to a pair of symmetric tasks. More concretely, the two tasks they focus on in this paper are NL-to-PL and PL-to-NL, and these assume that the natural language and code sufficiently and holistically capture one another. These assumptions would not be valid for certain types of tasks, such as summarization tasks, in which a very brief summary with a high-level overview is to be generated for a given input. In such cases, the summary would not sufficiently and holistically capture the input in such a way that something semantically equivalent to the input could be re-generated given only the summary.
- Based on my previous two points, it seems that the code summarization task (PL-to-NL) may not be symmetric to the (NL-to-PL) task. Namely, in $p_i \rightarrow n_{i+1}$, the model could generate a high-level natural language summary rather than a holistic description of $p_{i}$ with sufficient details for for generating $n_{i+1} \rightarrow p_{i+1}$ , with the expectation that $p_{i+1}$ would be semantically equivalent to  $p_{i}$. It is likely that the pretraining data for these code models include a lot of natural language with high-level descriptions of accompanying code (e.g., docstrings, NL StackOverflow test near code snippets). If this is the case, then the model is likely not great at generating holistic natural language descriptions for code. Therefore, it is possible that the model is very good at NL-to-PL (as per the high accuracy scores) but not as good at PL-to-NL, which results in low self-consistency scores. However, since it is not clear how to decouple this in the IdentityChain, it is difficult to clearly see this.

**Questions:**

1) Please comment on the points raised in the section above.
2) The results shown in Table 2 are unclear to me. Based on the description, it seems that the evaluation is on $sem(pl_0) - sem(nl_1)$. Therefore, it seems that each metric is computed based on $score(pl_0$, $nl_1)$. However, one of these is code and the other is natural language, so computing metrics like exact match on this would not make sense. Could you please clarify the details here?
3) In this framework, $pl_0$ is generated from a human-written specification $nl_0$. Since $nl_1$ is model-generated and not human-written, is it fair to expect that $nl_1 \rightarrow pl_1$ generates code that is similar to $pl_0$? Isn't it possible that the distributions are different?
4) In this paper, you focus on doing NL-to-PL first, which is natural since that is how HumanEval and MBPP are designed. Have you considered starting with PL-to-NL? Would you expect to see similar results?


Some minor comments:
- I would suggest re-considering calling this idea "self-consistency" as this term has been used to describe a different principle with LLMs: https://arxiv.org/abs/2203.11171.
- Section 5.3: "fin-grained" should be "fine-grained"
- Last line of page 6: "StarCoderBase-13B" is inconsistent with Table 1

---

> ### Author Response · Authors · 2023-11-11
> **Response to Reviewer AByS**
>
> Thank you for your valuable comments and feedback. We have updated the paper to fix typos and improve clarity. In addition, we hope to address your concerns and questions here:
>
> ## Weakness 1
> 1. We strongly agree that *"It is possible that the model is good at one and not the other"* and it is important to *"decouple these in their framework"*. In **Section 4.3**, we expressed our intention to build a holistic framework that evaluates the NL-to-PL Accuracy, PL-to-NL Accuracy, and Self-Consistency of a model separately at the same time.
>
> 2. Particularly, the metric $SSC_n$ which measures Strong Self-Consistency gives the model developers a summary of the model's overall performance. Then they can examine $Pass@1$ which measures NL-to-PL Accuracy, $SC_1$ which measures PL-to-NL Accuracy (explained in **Section 4.3, Paragraph 3**), $SC_n$ which measures Self-Consistney to determine which aspect is the weakest.
>
> 3. Following this line of reasoning, we evaluated these 4 aspects of the models and showed the results in **Table 1**. Comparing the $Pass@1$ score and the $SC_1$ score, we can tell whether a model is worse at NL-to-PL or PL-to-NL. For example, for **GPT-4**, the PL-to-NL Accuracy is higher. For **Code Llama-Instruct-7B**, they are about the same. For **GPT-3.5**, the PL-to-NL Accuracy is significantly lower.
>
>
> ## Weakness 2
> 1. This is a great point! We agree that not all pairs of tasks are symmetric and text expansion and summarization are not symmetric. In **Section 1, Paragraph 1**, we defined the task *"Code Summarization"* in a very specific way: the model is tasked to generate a natural language specification given a program instead of generating a high-level one-liner of what the program does.
>
> 2. In **Section 3.1, Paragraph 1**, we formalized the space $\mathcal{NL}$ as the space of all semantically valid and unambiguous program specifications in a specific natural language. Therefore, the NL-2-PL and PL-2-NL Generation in our setting are defined to be symmetric.
>
> 3. For our experiments, to enforce this symmetry, we carefully designed our [prompt template (run_identity_chain_openai.py, line 61)](https://github.com/anonymousauthor567/IdentityChain/blob/main/examples/run_identity_chain_openai.py) including emphasis and a one-shot example to make sure that the model is tasked to generate a detailed specification instead of a high-level one-liner. (**Appendix B**)
>
> ## Weakness 3
> To conclude, in our paper, **the models are tasked and enforced to generate detailed natural language specifications.** In this setting, our results showed that some models are worse at PL-to-NL Generation while some are worse at NL-to-PL Generation.
>
>
> ## Question 1
> 1. Hope our responses to the Weaknesses resolve your concern. Thanks a lot for the suggestions in *"Some minor comments:"*!
>
> ## Question 2
> 1. Thanks for asking. To evaluate the equality $sem(pl_0) = sem(nl_1)$, there are 3 approaches:
> - a. Human judgment, which directly compares $nl_1$ and $pl_0$, but it’s not scalable.
> - b. PL space automated metrics like CodeBLUE or TOM, which compares $pl_1$ and $pl_0$ to indicate the quality of the generation from $pl_0$ to $nl_1$.
> - c. NL space automated metrics like BLEU, which compares $nl_1$ and $nl_0$ to indicate the quality of the generation.
>
> 2. In this experiment, we first obtained the human-judged ground truth of $sem(pl0) = sem(nl1)$ and the results of some PL space and NL space metrics. We then calculate the correlations between the automated metric results and the human-judged ground truth. The results in **Table 2** showed that TOM score better correlates with the ground truth.
>
> ## Question 3
> 1. We expect that $pl_1$ should be semantically identical to $pl_0$ since the model should be able to understand its own output. This is why we consider **Self-Consistency Evaluation** to be fundamentally different from conventional consistency/robustness evaluation.
>
> 2. As for syntactical similarity, we do not assume anything. Depending on the model, $pl_1$ can be syntactically similar to or different from $pl_0$. As long as they are semantically identical, we consider the model to be self-consistent.
>
> ## Question 4
> 1. This is a great question! We mentioned in **Section 3.1, Last Paragraph** that our Self-Consistency definitions can be extended to starting with an arbitrary $pl_0$. However, as we want to evaluate self-consistency in the PL space, there is a natural asymmetry regarding the starting point.
>
> 2. Particularly, if we start with a human-crafted $pl_0$, generate the chain $pl_0 \to nl_0 \to pl_1 \to ...$, and evaluate in PL space, we can still evaluate Strong Self-Consistency but not (Weak) Self-Consistency w.r.t $pl_0$, since $sem(nl_0) = sem(pl_1)$ can't be automatically measured.
>
> ## Thank You
> Please let us know if the above clarification resolves your concerns. If there are any further questions, we'll be more than pleased to elaborate more.
>
> Thanks again for your time and consideration.

---

> > ### Comment · Reviewer_AByS · 2023-11-21
> >
> > Thank you for the detailed answers.
> >
> > For Weakness 1, it is not clear to me how it is possible to directly compare the $pass@1$ and $SC_1$ scores. These are two different metrics, measuring very different things.
> >
> > Also, just as a small note, even if this was valid, it seems that the statement "For GPT-3.5, the PL-to-NL Accuracy is significantly lower" is false because the $SC_1$ for MBPP is 86.4 but $Pass@1$ is 68.9, so by that definition, the PL-to-NL Accuracy is significantly higher. For HumanEvalPlus, it is the opposite.

---

> ### Author Response · Authors · 2023-11-21
> **Response to Reviewer AByS**
>
> Thank you for the reply. We would like to further clarify on **Weakness 1**.
>
> For the statement *"For GPT-3.5, the PL-to-NL Accuracy is significantly lower"*, we agree that the performance is not consistent across MBPP and HumanEvalPlus. However, we mainly referred to the results of HumanEvalPlus because it is a more accurate benchmark with **16.5** test cases per task, while MBPP only has **3** test cases per task (**Section 5**). This is also the reason why we chose HumanEvalPlus instead of the original HumanEval since the test case coverage is significantly enhanced.
>
> For the comparison between $Pass@1$ and $SC_1$, in our framework, they are both measuring accuracy (**Section 4.1, Section 6.4**). In a chain $nl_0 \to pl_0 \to nl_1 \to ...$, the $Pass@1$ score calculates the percentage of $pl_0$ that passes all test cases, which is NL-to-PL Accuracy. As defined in **Section 3.2**, the $SC_1$ score calculates the percentage of $nl_1$ that is semantically consistent with $pl_0$, which is PL-to-NL Accuracy.
>
> Indeed, these two scores separately measure the model's ability of NL-to-PL and PL-to-NL Generation, and they won’t be comparable if they are not both measuring accuracy. However, in our framework, $Pass@1$ and $SC_1$ are both measuring accuracy, which are absolute measurements *i.e.* how far away the model’s current performance is from absolute correctness. In this sense, we can compare them to see if the model is further from absolute correctness regarding NL-to-PL Generation or PL-to-NL Generation.
>
> Currently, no metric can directly measure the semantic consistency between $nl_1$ and $pl_0$. Therefore, as discussed in **Section 4.1**, we design a relaxed accuracy metric **TOM** score as an implementation of the ideal $SC_1$ score and show in **Section 6.2** that **TOM** score correlates better with ground truth than other existing metrics. We recognize that there will be better metrics than **TOM** score in the future, but within our framework, the definition of $SC_1$ is separated from the implementation. Therefore, future work can simply replace **TOM** score with a better metric as an improvement of the IdentityChain framework.

---

> > ### Comment · Reviewer_AByS · 2023-11-22
> >
> > Thank you for the clarification. I have read the other reviews and responses. I have increased my score.

---

> ### Author Response · Authors · 2023-11-22
> **Thank You Note**
>
> We really appreciate your precise and thoughtful comments, which helped us improve the clarity of our presentation.
>
> Thanks again for the constructive feedback and support for our work!

---

### Official Review · Reviewer_6ee8 · 2023-10-30

**Soundness:** 3 good
**Presentation:** 3 good
**Contribution:** 1 poor
**Rating:** 3
**Confidence:** 3

**Summary:**

The paper proposes to evaluate code LLMs with respect to self-consistency (including documentation for code and code). To this end they perform a formal definition and evaluate multiple models. The paper shows that self-consistency is a concerning aspect.

This behavior is highly anticipated as it is known for LLM in general. Thus, the research question, while worth asking, is already quite likely to yield only limited insights. The authors would have to motivate much more, why different behavior could be anticipated and then carefully analyze to what extent this reasoning holds (or does not). In the intro the paper does present first steps in this direction though lacking reasoning, i.e., the mere fact that a docstring does not allow to generate code but a natural language completion does is interesting, but per se not surprising. Maybe, it is due to training data? Maybe due to sth else? The authors should carefully think about such questions. In the analysis some findings are not sufficiently explained or might be spurious or follow general known patterns ("LLMs don't understand (code) semantics"). A positive example is the lack of awareness of data types, which is specific to code. Finding multiple specific aspects and then generalizing to known model behavior facts is significantly more interesting as it provides deeper insights. In summary, the paper lacks depth and addresses a question that from the outset is of some but not much interest given existing knowledge on LLMs.

Detailed comments:
*  For example, we observe that stacking more parameters does not necessarily guarantee the improvement of self-consistency: while StarCoderBase-7B performs worse in Pass@1 for both benchmarks than StarCoderBase-13B, the former outperforms the latter after five self-iterations
 This is a single example, no statistical analysis. Also in general large models seem to perform better. So this finding is of very limited relevance. Looking at performance of LLMs there is often some variation.

**Strengths:**

see above

**Weaknesses:**

see above

**Questions:**

see above

---

> ### Comment · Reviewer_6ee8 · 2023-11-21
>
> The authors did not provide a rebuttal.

---

> ### Author Response · Authors · 2023-11-21
> **Response to Reviewer 6ee8**
>
> Thank you for your comments and feedback. We have updated the paper to fix typos and improve clarity. In addition, we hope to address your concerns and questions here:
>
> ## Weakness 1: Not Enough Contribution
> 1. The consistency issue of LLMs has been identified by many previous works as we summarized in related work **Section 2, Paragraph 2**. However, as highlighted in **Section 1** of the updated version, **"Self-Consistency is different from Consistency"**. Compared to conventional consistency, self-consistency is a different property of trustworthy models since self-consistency violations reveal that **the model doesn't even understand its own output**. Conventional consistency evaluations transform the inputs using some human-crafted perturbations and then test the drop in accuracy of the models. It is known, to some extent, that a model doesn't understand some human-crafted transformations since it might not have seen similar training data. However, it's a more serious issue that the model doesn't understand itself.
>
> 2. To our best knowledge, this **Self-Consistency** property is not well-explored even for general LLMs across NLP tasks (**Section 2, Paragraph 2**). As we illustrate in our Future Work Section (**Appendix A**), it is indeed worthwhile to answer *"why self-consistency violation happens and how to fix it"*. To do so, however, it is important to first properly state the question and formalize the problem. In this paper, we identify and formalize the concept of Code LLMs' self-consistency across two major open-ended generation tasks: NL-to-PL and PL-to-NL Generation.
>
> 3. We want to emphasize that **self-consistency is different from accuracy and should be taken into account during evaluation**. This insight might seem intuitive, but it doesn't mean that researchers and model developers don't need to hear it. To our best knowledge, even the most recent Code LLMs are evaluated solely for their accuracy when published. As indicated in our title **Beyond Accuracy**, we encourage model developers and researchers to evaluate and improve self-consistency in addition to accuracy by not only demonstrating severe self-consistency violations but also **providing them with an effective and efficient evaluation tool**, the IdentityChain, as we propose in **Section 6**.
>
> 4. We are curious about *"why self-consistency violation happens and how to fix it"*. However, this won't be possible if we don't even have a tool to measure self-consistency. As mentioned in **Section 4.3**, we propose a holistic evaluation framework that evaluates the NL-to-PL Accuracy, PL-to-NL Accuracy, and Self-Consistency at the same time. The model developers can use our IdentityChain framework to pinpoint the weaknesses of their models. Furthermore, we demonstrated in **Section 6.4** with 3 specific weaknesses identified using IdentityChain (**Figure 6,7,8**).
>
> 5. To conclude, in this paper, we identify and formalize a new property, self-consistency of Code LLMs. We then build an effective and efficient tool to evaluate this property. Moreover, this tool we built empowers further exploration of more complex questions like *"why self-consistency violation happens and how to fix it"*.
>
>
> ## Weakness 2: *"some findings are not sufficiently explained or might be spurious or follow general known patterns"*
>
> Thanks for letting us know. It would be much appreciated if you could help point out which particular findings are referred to. We will try our best to provide better explanations or to fix potential mistakes.
>
>
> ## Detailed Comment
> 1. For the StarCoder 7B versus 15B case (should be 15B instead of 13B, we apologize for the typo), it is a statistical analysis. Both models are evaluated on 2 benchmarks and each of the benchmarks contains about 200 samples (**Section 5**). This is a controlled experiment since to our best knowledge of the training process of StarCoder, the only difference between the two models is the size.
>
> 2. In addition, we have shown in **Section 6.3** that the $SCC_n$ and $SC_n$ scores of StarCoder reach a fixed point within 5 steps. Therefore, the chain length does not affect the results.
>
> 3. Our observation is that *"stacking more parameters does not necessarily guarantee the improvement of self-consistency"*, which means that the claim *"stacking more parameters guarantees the improvement of self-consistency"* is not true. The StarCoder case provides exactly a counterexample to disprove this claim.
>
> 4. The key insights drawn from this observation is that **self-consistency is different from accuracy** and we need **more than merely *"stacking more parameters"* to improve the self-consistency of Code LLMs.**
>
> ## Thank You
> Please let us know if the above clarification resolves your concerns. If there are any further questions, we'll be more than pleased to elaborate more.
>
> Thanks again for your time and consideration.

---

> ### Author Response · Authors · 2023-11-22
> **Follow-up on Response to Reviewer 6ee8**
>
> Dear Reviewer 6ee8,
>
> In light of the updates in the paper and clarifications provided in our previous response, we would be more than grateful if you could comment on whether the concerns and questions in the initial review are addressed. We will try our best to provide more detailed explanations if needed.
>
> Thanks again for your time and consideration.

---

### Official Review · Reviewer_d4Cs · 2023-10-31

**Soundness:** 3 good
**Presentation:** 3 good
**Contribution:** 4 excellent
**Rating:** 8
**Confidence:** 3

**Summary:**

The paper focuses on evaluating the self-consistency of Code Large Language Models (Code LLMs). The authors stress the importance of self-consistency in these models as they might generate varying outputs for semantically similar inputs. They introduce an evaluation framework, IdentityChain, which helps in understanding the model's vulnerabilities and improves the architecture of Code LLMs. Furthermore, they discuss various existing evaluation metrics for Code LLMs, emphasizing that many of these do not consider the model's self-consistency. By using their framework, they aim to provide better insight into the models and improve the training or architectural design of Code LLMs in the future.

**Strengths:**

Originality: The paper addresses the lesser-explored topic of self-consistency in Code LLMs. The introduction of the IdentityChain framework showcases an original approach to understanding the internal workings and vulnerabilities of such models.

Quality: The paper cites multiple sources and provides a comprehensive review of the existing literature, giving it a solid foundation. The work is well-researched and backed by evidence.

Clarity: The paper is structured logically, with a clear progression from the introduction of the problem to the presentation of the solution. The content is presented in an organized manner, making it relatively easy to understand for readers.

Significance: Addressing self-consistency in Code LLMs has potential implications for the future development and deployment of such models, making the research both timely and significant.

**Weaknesses:**

Generalization: The paper specifically discusses Code LLMs, and it would be useful to understand how the principles and findings apply to other LLMs or different types of models.

Comparison with Existing Solutions: While the paper reviews existing evaluation metrics, a direct comparison in terms of performance or effectiveness with the proposed solution might have added more clarity.

**Questions:**

How does the IdentityChain framework compare in terms of efficiency and accuracy with other existing evaluation tools for Code LLMs?
Are there any plans to extend the research to encompass other types of LLMs or machine learning models?
How can the IdentityChain framework be integrated into the current training methodologies of Code LLMs for better results?

---

> ### Author Response · Authors · 2023-11-11
> **Response to Reviewer d4Cs**
>
> Thank you for your valuable comments and feedback. We have updated the paper to fix typos and improve clarity. In addition, we hope to address your concerns and questions here:
>
>
> ## Weakness 1
> 1. Thanks for pointing this out. Generalization is what we plan to do as an immediate next step. We added a Future Work Section in **Appendix A**, mentioning that we can easily incorporate PL-to-PL and NL-to-NL Generation into the IdentityChain framework, and we plan to do that as one of our immediate next steps. We believe that IdentityChain can provide a better effective metric for NL-to-NL Generation (translating docstrings from one language to another) leveraging our proposed dynamic metric, TOM score.
>
> 2. For different models like RNN or State-Space Models, as long as they are pre-trained on code and fine-tuned on NL-to-PL and PL-to-NL Generation, the IdentityChain framework is directly applicable since it has no restriction on the model architecture.
>
> 3. We recognize that our IdentityChain framework is hard to generalize to models NOT pre-trained on code. In theory, the definitions of **Self-Consistency** and **Strong Self-Consistency** can be extended to any symmetric tasks. In practice, however, there are currently no good metrics to evaluate self-consistency for symmetric tasks like NL-to-Vision and Vision-to-NL Generation. The essence of our method is to **map natural language problems to the program space and evaluate them more accurately with test cases**. This is also one of the implicit insights we want to provide to the readers: **mapping reasoning problems in other modalities (Natural Language Inference, Visual Reasoning, etc.) to code is a promising future direction**.
>
> 4. We also recognize that our current definition of self-consistency requires the tasks to have a semantically symmetric nature, but the definitions can be extended to further incorporate tasks with semantic entailment relationships. For example, if a model is capable of both Bug Detection and Bug Fixing, then it should be self-consistency across detection and fixing.
>
>
> ## Weakness 2
> 1. Thanks for asking. The direct competitors of our evaluation metric TOM score, are PL space metrics like CodeBLEU and NL space metrics like BLEU or BERTScore. Therefore, in **Section 6.2**, we discussed an experiment comparing the effectiveness of TOM score to that of the competitors, and show the results in **Table 2**.
>
> 2. Specifically, we first obtained the human-judged ground truth of $sem(pl0) = sem(nl1)$ and the results of some PL space and NL space automated metrics. We then calculate the correlations between the automated metric results and the human-judged ground truth. We show through results in **Table 2** that TOM score better correlates with the ground truth than existing PL space and NL space metrics.
>
>
> ## Question 1
> 1. This is a great question! To our best knowledge, we haven't found existing work that does similar self-consistency evaluations that we can directly compare to.
>
> 2. In terms of accuracy, we compared our proposed metric TOM score with some popular PL space and NL space metrics, as elaborated in our response to Weakness 2.
>
> 3. In terms of efficiency, we reported absolute results in **Section 6.3** showing that by leveraging greedy decoding, most of the non-self-consistency cases can be revealed within the first 3 steps, even though we chose 5 steps for all our experiments. Therefore, assuming that the model's inference time is $t$ for both NL-to-PL and PL-to-NL Generation, empirically, a 3-step IdentityChain evaluation using greedy decoding takes no more than $7t = t + 3 * 2t$.
>
>
> ## Question 2
> 1. We hope that our response to Weakness 1 to some extent answers this question.
>
> 2. We do plan to explore more on how to **map reasoning problems in other modalities to code** as one of the next steps. However, we have to recognize that this is an open and challenging problem.
>
>
> ## Question 3
> 1. This is also a very pertinent question! In the Future Work Section in **Appendix A**, we discuss more about how to improve current Code LLMs using IdentityChain.
>
> 2. An immediate downstream application of IdentityChain is **Distillation**. We can use IdentityChain to synthesize self-consistent parallel data generated by more capable models like GPT-4, and use them to fine-tune small models, aiming for self-consistency improvements.
>
>
> ## Thank You
> Please let us know if the above clarification resolves your concerns. If there are any further questions, we'll be more than pleased to elaborate more.
>
> Thanks again for your time and consideration.

---

> ### Author Response · Authors · 2023-11-22
> **Follow-up on Response to Reviewer d4Cs**
>
> Dear Reviewer d4Cs,
>
> Thank you for your positive feedback and support for our work! In light of the updates in the paper and clarifications provided in our previous response, we would be more than grateful if you could comment on whether the questions in the initial review are addressed. We will try our best to provide more detailed explanations if needed.
>
>
> Thanks again for your time and consideration.

---

### Official Review · Reviewer_G8dX · 2023-11-03

**Soundness:** 2 fair
**Presentation:** 3 good
**Contribution:** 2 fair
**Rating:** 6
**Confidence:** 4

**Summary:**

This paper presents a concept of self-consistency for Code LLMs and proposes a framework, namely IdentityChain, to evaluate their self-consistency. Using IdentityChain, the paper studies the self-consistency of a series of CLMs beyond the accuracy based on the HumanEval+ and MBPP benchmarks and reveals some inconsistent behaviors of those models.

**Strengths:**

Leveraging unique properties of code such as executability and dynamic engine feedback, this paper formalizes a definition of self-consistency for CLMs. The consistency is interpreted on the NL-PL-NL chain, unlike recent work on self-consistency such as Elazar et al. (2021), which defines the consistency as invariance of LLMs to semantic preserving transformations of the prompts.

The authors propose IdentityChain, a framework for self-consistency evaluation with a new metric, named Test Output Match (TOM) score. TOM is based on matching binary outcome of dynamic test execution as well as syntax and runtime errors.

**Weaknesses:**

There are some weaknesses in the self-consistency formulation:

1. We could construct a perfectly self-consistent and not-so-useful CLM. For example, a CLM could generate summarization nl_1 exactly the same as pl_0. It is likely that the model generates pl_1 semantically equivalent to the input pl_0 in the next step. The chain goes on and the model is mostly perfectly self-consistent. Based on that, we could have very high self-consistency but that does not say much about the model.

2. An important aspect of consistency is the consistent behavior to transformations of the prompts with equivalent semantics. This paper does not touch upon it.

3. While the concept and the proposed framework sound general to any task, it seems to only apply to code synthesis and summarization. What about other tasks for code such as reasoning about code, fixing bugs?

**Questions:**

At the end of Page 2, it is said that “the underlying semantics of pl1, nl1, pl0 should all be the same. We call such a behavioral property self-consistency”, but in the next paragraph, the authors write “a model’s accuracy is low, its self-consistency can remain high when the model, despite making initial errors, consistently maintains those errors”. Please clarify.

“it relatively improves GPT-3.5 by 57.5% in FPS-5” What is FPS-5 here?

---

> ### Author Response · Authors · 2023-11-11
> **Response to Reviewer G8dX**
>
> Thank you for your valuable comments and feedback. We have updated the paper to fix typos and improve clarity. In addition, we hope to address your concerns and questions here:
>
>
> ## Weakness 1
> 1. We would appreciate a clarification on "*CLM could generate summarization nl_1 exactly the same as pl_0*". Does it mean that the model copy-pastes $pl_0$ as $nl_1$? If so, this model is not performing PL-to-NL Generation and therefore out of the scope of self-consistency evaluation by our definition. For our experiments, we carefully designed our [prompt template (run_identity_chain_openai.py, line 61)](https://github.com/anonymousauthor567/IdentityChain/blob/main/examples/run_identity_chain_openai.py) including a one-shot example to make sure that the model actually generates a natural language specification.
>
> 2. We agree that *"We could construct a perfectly self-consistent and not-so-useful CLM."*. In **Section 4.3, Paragraph 4**, we mentioned that *"An ideal model should be both accurate and self-consistent.
> An accurate but not self-consistent model is not trustworthy, while a self-consistent but not accurate model is useless."* This is particularly why we defined the concept of **Strong Self-Consistency** in **Section 3.1, Definition 2**. An ideal model should be strong self-consistent, which is both accurate and self-consistent.
>
> 3. More importantly, we discuss in **Section 4.3** and show through results in **Table 1** that IdentityChain is a holistic evaluation framework that efficiently evaluates NL-to-PL Accuracy, PL-to-NL Accuracy, and Self-Consistency at the same time. In **Section 6.4**, we further discuss that model developers can use IdentityChain to pinpoint the weaknesses of their models.
>
>
> ## Weakness 2
> 1. We agree that *"An important aspect of consistency is the consistent behavior to transformations of the prompts with equivalent semantics."* We highlight in **Section 1** of the updated version that **"Self-Consistency is different from Consistency"**. We also discussed in the related work **Section 2, Paragraph 2** that conventional consistency violation is different from self-consistency violations.
>
> 2. We believe that self-consistency is a different property of a trustworthy model since self-consistency violations reveal that **the model doesn't even understand its own output**. However, conventional consistency evaluation settings involve human-crafted transformations. It is to some extent acceptable if a model doesn't understand some human-crafted transformations since it might have not seen similar training data, but it's a more serious issue that the model doesn't understand itself.
>
> 3. More importantly, in an IdentityChain $nl_0 \to pl_0 \to nl_1 \to ...$ as illustrated in **Figure 1**, $nl_1$ is in fact a **semantic-preserving transformation** of $nl_0$, which is difficult to obtain using rule-based human-crafted transformations.
>
> ## Weakness 3
> 1. Thank you for asking! We added a Future Work Section in **Appendix A**, mentioning that it's easy to incorporate PL-to-PL and NL-to-NL Generation into our framework, and we plan to do that as an immediate next step.
>
> 2. For reasoning tasks like Vulnerability Detection, conceptually they can be evaluated together with PL-to-PL Generation following the setups in this [related work](https://arxiv.org/abs/2305.11662), which asks a model to translate a QA task from English to another language and ask the same model to answer the QA task in both languages.
>
> 3. We recognize that our current definition of self-consistency requires tasks with a symmetric nature, but the definitions can be extended to further incorporate tasks with semantic entailment relationships. For example, if a model is capable of both Bug Detection and Bug Fixing, then it should be self-consistency across detection and fixing.
>
>
> ## Question 1
> 1. By *“the underlying semantics of pl1, nl1, pl0 should all be the same. We call such a behavioral property self-consistency”*, we meant that **Self-Consistency** doesn't require the initial generation from $nl_0$ to $pl_0$ to be accurate. As long as the semantics of $pl_0$, $nl_1$, and $pl_1$ are the same, the model is self-consistent.
>
> 2. In contrast, **Strong Self-Consistency** requires the semantics of $nl_0$ to be the same as $pl_0$, in addition to being self-consistent. Therefore, we wrote that *“a model’s accuracy is low, its self-consistency can remain high when the model, despite making initial errors, consistently maintains those errors”.*
>
> 3. **"Self-Consistency is different from Accuracy"** is one of the key insights that we want to provide to the readers. Ultimately, we want a model to be both self-consistency and accurate, as elaborated in our response to Weakness 1.
>
>
> ## Question 2
> 1. Thank you for pointing that out. It is fixed in the updated version.
>
>
> ## Thank You
> Please let us know if the above clarification resolves your concerns. We'd be more than pleased to elaborate more. Thanks again for your time and consideration.

---

> ### Author Response · Authors · 2023-11-22
> **Follow-up on Response to Reviewer G8dX**
>
> Dear Reviewer G8dX,
>
> In light of the updates in the paper and clarifications provided in our previous response, we would be more than grateful if you could comment on whether the concerns and questions in the initial review are addressed. We will try our best to provide more detailed explanations if needed.
>
> Thank you for your time and consideration.

---

> ### Comment · Reviewer_G8dX · 2023-11-22
>
> Thank the authors for the detailed response.
>
> I still think that a highly accurate model can be perfectly self-consistent (based on the definition proposed in this work) without actually preserving the original semantics of $nl_0$ in the chain, so there are cases where this self-consistency is not helpful. However, I can see some value of the proposed framework. Therefore, I increase my score a little bit.

---

> > ### Author Response · Authors · 2023-11-22
> > **Thank You Note**
> >
> > We really appreciate your prompt response and the question you raised. Though currently, we cannot think of a concrete non-trivial example where *"self-consistency is not helpful"*, it is a worthwhile question for us to explore further.
> >
> > Thanks again for your constructive feedback and openness to our work!

---

### Author Response · Authors · 2023-11-23
**Summary of the Discussion Stage**

We thank all reviewers for their feedback and engagement during the discussion period. In addition, we would like to briefly summarize the resolved major concerns and the values we hope to provide to the entire ICLR community via this paper.


## Resolved Concerns

1. **How is self-consistency different from consistency?** We clarified that self-consistency is a different property from conventional consistency since self-consistency violations reveal that the model doesn't even understand its own output. On the other hand, conventional consistency evaluations transform the inputs using some human-crafted perturbations and then test the drop in accuracy of the models. It is understandable, to some extent, that a model doesn't understand specific human-crafted transformations since it might not have seen similar training data. However, it's a more serious issue that the model doesn't understand itself.

2. **How to compare the model's performance on NL-to-PL Generation v.s. PL-to-NL Generation in our framework?** We clarified that since the metrics in our IdentityChain framework, both NL-to-PL metric($Pass@1$) and PL-to-NL metric ($SC_1$) are measuring accuracy, which are absolute measures. In that sense, we can compare them to see which accuracy is further from absolute correctness.

3. **How to generalize the concept of self-consistency to more tasks?** We clarified that our IdentityChain framework is by design compatible with symmetric tasks like NL-to-NL and PL-to-PL Generation. Moreover, we can extend our definition of self-consistency to tasks with entailment relationships. For example, a trustworthy model should be self-consistent across Bug Detection and Bug Fixing *i.e.* fixing a bug should entail that a bug exists.


## Values to ICLR Community

1. **We provide the evidence and the tool to convince researchers and model developers in the ICLR community to evaluate their models "beyond accuracy" and take a new property, self-consistency, into account.** In this paper, we empirically show that self-consistency violations imply significant trustworthiness concerns and conventional accuracy does not imply self-consistency. However, to our best knowledge, even the latest Code LLMs are still only evaluated on their conventional accuracy. To raise awareness of this critical existing issue, we first show through experiments that current models suffer from severe self-consistency violations. More importantly, we design an effective and efficient tool, the IdentityChain framework, for researchers and developers to conveniently evaluate the self-consistency and conventional accuracy of their own models at the same time (**Section 4.3**).

2. **We provide the building blocks for researchers and model developers in the ICLR community to study valuable questions like "Why do self-consistency violations happen? How do we improve self-consistency?"** In this paper, we formalize this new self-consistency property and build a tool for self-consistency evaluation. More importantly, we demonstrate the potential of our IdentityChain framework as a model debugging tool to help developers pinpoint and study the weaknesses of their models. As discussed in **Section 6.4** and **Appendix A**, hopefully, our framework and exploratory studies can inspire new training methods or new model architectures that improve self-consistency without sacrificing conventional accuracy.

---

### Meta-Review · Area_Chair_iGYy · 2023-12-06

**Metareview:**

This paper proposes the idea of self-consistency as an additional "dimension" for evaluating Code LLMs. The idea is that a model "should be self-consistent when generating documentation for its own code and generating code for its own natural language specifications". The authors evaluate the self-consistency of multiple code LLMs and discuss the limitations of existing models.


## Strengths
- A simple and clear idea.
- A good set of evaluations.
- Self-consistency seems like a good tool for detecting error modes in LLMs (e.g. as in Sec 6.4)

## Weaknesses
- It is unclear if self-consistency provides additional information wrt to the abilities a Code LLM, i.e., there is no evidence to suggest that self-consistency doesn't (significantly) correlate with task accuracy (in the "traditional" sense, e.g. as in pass@k for synthesis). If accuracy and self-consistency correlated well, then one of them could be redundant as a measure of model quality.
- Code summarization is a lossy step "by design":  a good summary may reasonably omit details/corner-cases and how these are concretely handled. In that sense, it's unclear if a model that is good at summarizing code (especially real-life code and not like the relatively simple snippets contained in HumanEval and MBPP) should actually achieve a high self-consistency score. If a summary is _not_ lossy, then it risks being a bad summary.
- The evaluation is only performed in introductory-level programming snippets.

Given the above, I would recommend this paper to be accepted and kindly ask the authors to incorporate the reviewer and meta-review feedback in their work.

**Justification For Why Not Higher Score:**

The "weakness" mentioned in the main review, suggests that while this is an interesting idea, it might not be a major contribution.

**Justification For Why Not Lower Score:**

This is a good, technically correct paper. There is no reason to reject it.

---

### Decision · Program_Chairs · 2024-01-16

Accept (poster)